# High-Dose Intravenous Steroid Treatment Seems to Have No Long-Term Negative Effect on Bone Mineral Density of Young and Newly Diagnosed Multiple Sclerosis Patients: A Pilot Study

**DOI:** 10.3390/biomedicines11020603

**Published:** 2023-02-17

**Authors:** George Simeakis, Maria Anagnostouli, Nikolaos Fakas, John Koutsikos, Athanasios Papatheodorou, Konstantinos Chanopoulos, Kwnstantinos Athanasiou, George Papatheodorou, Evangelia Zapanti, Maria Alevizaki, Gregory Kaltsas, Evangelos Terpos

**Affiliations:** 1Endocrine Department, 401 General Military Hospital of Athens, 115 25 Athens, Greece; 2Department of Clinical Therapeutics, School of Medicine, National and Kapodistrian University of Athens, 115 28 Athens, Greece; 3Multiple Sclerosis and Demyelinating Diseases Unit, 1st Neurology Department, School of Medicine, Aeginition University Hospital, National and Kapodistrian University of Athens, 115 28 Athens, Greece; 4Neurology Department, 401 General Military Hospital of Athens, 115 25 Athens, Greece; 5Department of Nuclear Medicine, 401 Military Hospital of Athens, 115 25 Athens, Greece; 6Department for Biomedical Research, 251 Air Force General Hospital, 115 25 Athens, Greece; 7Center for Molecular Biology—Research Unit, 401 Military Hospital of Athens, 115 25 Athens, Greece; 8Endocrine Oncology Unit, First Department of Propaedeutic and Internal Medicine, Laiko Hospital, National and Kapodistrian University of Athens, 115 27 Athens, Greece

**Keywords:** multiple sclerosis, glucocorticoids, BMD, cytokines, periostin, 25-hydroxyvitamin D, bone-turnover, osteoporosis

## Abstract

High-dose intravenous steroid treatment (HDIST) represents the first choice of treatment for multiple sclerosis (MS) relapses. Chronic oral glucocorticoid (GC) administration correlates with bone loss whereas data regarding HDIST in MS are still conflicting. Twenty-five newly diagnosed MS patients (NDMSP) (median age: 37 years) were prospectively studied for the effects of HDIST on bone mineral density (BMD) and bone metabolism. Patients received 1000 mg methylprednisolone intravenously every day for 5 days followed by oral prednisolone tapering over 21 days. Bone metabolism indices were determined prior to GC, on days 2, 4, 6, and 90, and at months 6, 12, 18, and 24 post GC therapy. Femoral, lumbar-spine BMD, and whole-body measurement of adipose/lean tissue were assessed prior to GC-administration and then every six months. Ten patients completed the study. N-terminal-propeptide-procollagen-type-1 and bone-specific alkaline phosphatase showed a significant increase at day-90 (*p* < 0.05). A transient non-significant fall of BMD was observed at 6 months after GC-administration, which subsequently appeared to be restored. We conclude that HDIST seems not to have long-term negative effects on BMD, while the observed transient increase of bone formation markers probably indicates a high bone turnover phase to GC-administration. Additional prospective studies with larger sample size are needed.

## 1. Introduction

Multiple sclerosis (MS) is a chronic autoimmune inflammatory demyelinating and neurodegenerative disease of the central nervous system (CNS), occurring mainly in young patients (20–40 years old). It is the most common cause of non-traumatic disability in young adults while the clinical spectrum of the disease varies depending on the location and extent of demyelinating lesions [1,2,3,4]. Assessment of neurological impairment according to the severity of CNS signs and symptoms is performed with the extended disability status scale (EDSS) [5]. The EDSS score ranges from 0 to 10 in 0.5-unit increments; zero indicates absence of any neurological findings (i.e., no disability in any functional system (FS)) and ten indicates death due to MS.

High-dose intravenous steroid treatment (HDIST) represents the first choice of treatment for MS relapses [1]. Glucocorticoids (GCs) act on the immune system by inhibiting proinflammatory cytokines, suppressing the activity of Type 1 T helper (Th1) lymphocytes and downregulating the expression of the major histocompatibility complex (MHC) molecules; HDIST induces the process of programmed T cell death with the effect lasting for at least six months after GC administration [6]. Although chronic oral GC administration has been associated with bone loss, data regarding the effects of HDIST on bone mineral density (BMD) and bone metabolism in MS patients are still conflicting [7,8,9]. GC-induced osteoporosis is the most common form of secondary osteoporosis and may lead to increased incidence of osteoporotic fractures with the vertebrae being the most frequent site [10,11,12,13]. Pathophysiology and related mechanisms which mediate the GC effect on bone remodeling include decreased bone formation through Wnt signaling inhibition and increased bone resorption through the dysregulation of the receptor activator of nuclear factor kappa-Β ligand (RANKL)/osteoprotegerin (OPG) system leading to bone loss; an illustrative overview of GC action on bone remodeling is depicted in Figure 1. Although the mechanisms involved in GC-induced osteoporosis have extensively been studied, there is substantial heterogeneity of the type and the extent of associated skeletal adverse effects [14,15]. In addition, although there is a clear correlation between chronic, oral GC administration, and bone loss, similar data regarding the intravenous administration of high GC doses in MS relapses and other autoimmune diseases along with changes in Bone Turnover Markers (BTMs) are contradictory. A possible plausible explanation could be the coexistence of other osteoporosis risk factors in this subgroup of patients. Inflammatory cytokines, such as Interleukin (IL)-1, IL-6, and IL-17, which are involved in the pathogenesis of many autoimmune diseases including MS, enhance differentiation and activation of osteoclasts, posing a possible indirect effect in bone loss mediated by RANKL. Interestingly, an increased incidence of osteoporosis and hip fractures is observed in MS patients, even during the early stages of the disease, indicating that factors other than cumulative corticosteroid dose may predispose such patients to bone loss [16,17,18,19,20,21]. Although several studies have attempted to determine the role of HDIST in bone loss specifically in MS patients, no clear-cut conclusions have been drawn so far and therefore no guidelines have been established for the prevention and/or treatment of bone loss in these patients [22,23].

Regarding BTMs and their correlation with the changes of bone mass (BM), there is evidence that can be used to monitor bone metabolism in patients with autoimmune diseases under GC treatment [24,25,26]. Moreover, BTMs may serve as prognostic markers for GC induced bone loss assessment [24] as well as for monitoring the response to osteoporosis treatment [27].

The aims of this pilot study were to prospectively investigate the effect of HDIST on bone metabolism in newly diagnosed MS patients (NDMSP) and evaluate the relationship between BM changes, BTMs, and inflammatory cytokines.

## 2. Materials and Methods

### 2.1. Subjects

Twenty-five NDMSP (10 women) with previous minor symptoms, mainly sensory disturbances, were referred to the Neurology Department of the 401 Military Hospital of Athens and prospectively enrolled during a period of 5 years; all patients met the 2017 McDonald Criteria for MS diagnosis [28]. They received 1000 mg of Methylprednisolone intravenously every day for 5 consecutive days, followed by 60 mg of oral prednisolone per day, the dose being tapered over 21 days. Main inclusion criteria also included: age 18–45 years, fully ambulatory patients (EDSS ≤ 1), and women with regular menstruation periods. Exclusion criteria were defined as: medical history of any chronic disease, previous GC treatment in any dosage regimen, treatment with agents affecting bone metabolism, and EDSS ≥ 2 during follow-up. Seven patients were excluded due to mobility impairment, with an increase in EDSS score (EDSS ≥ 2) in the first 3 months of follow-up, and a further eight were lost to follow up. In the remaining 10 patients (6 men), demographic and clinical data, including disease modifying treatments (DMTs), were evaluated prior to GC administration and then on days 2, 4, 6, and 90, and months 6, 12, 18, and 24. Clinical activity of the disease was assessed by EDSS score and brain/spine magnetic resonance imaging (MRI) at baseline, 90 days post diagnosis, and then every 6 months until the end of the study. During the 24-month follow-up, DMT was initiated to all patients (interferon beta n = 4; fingolimod n = 2; natalizumab n = 1; dimethyl fumarate n = 1; teriflunomide n = 1; cyclophosphamide n = 1). None of the patients received additional glucocorticoid treatment during the follow-up due to the absence of any clinical or neuroimaging activity, while all patients received oral active vitamin D to maintain serum levels of 30–40 ng/mL.

The study protocol was approved by the local institutional ethics committee of National and Kapodistrian University of Athens—Alexandra Hospital and 401 General Military Hospital of Athens while all subjects provided written, informed consent before recruitment according to the Declaration of Helsinki.

### 2.2. Bone Mineral Densitometry

BMD of both hips and lumbar spine as well as whole-body measurement of adipose and lean tissue were assessed with dual-energy X-ray absorptiometry (DXA) (LUNAR Prodigy, GE Medical Systems) prior to GC administration and consecutively every six months. All DXA measurements were performed by the same experienced operator.

### 2.3. Biochemistry

Measurements of the biological variables were made on blood samples collected between 8:00 and 9:00 A.M. after an overnight fast prior to GC administration and consecutively on days: 2, 4, 6, and 90, and months: 6, 12, 18, and 24. Serum samples were stored frozen at −80 °C until assayed. Serum levels of calcium, phosphorus, albumin, magnesium, and creatinine were measured using the I Lab Taurus—Clinical Chemistry Analyzer; total 25-hydroxyvitamin D (25OHD) was measured by Roche Elecsys Vitamin D total assay; intact parathormone (iPTH) by Biomerica Intact-PTH ELISA and thyroid hormones (TH), follicle stimulation hormone (FSH), luteinizing hormone (LH), total testosterone (Testo), prolactin (PRL), and estradiol (E2) by Beckman Coulter RIA/IRMA KIT, according to standard laboratory protocols. IL-1β, IL-6, and IL-17 were measured by Human IL Immunoassay ELISA, R&D Systems, with intra- and inter-assay coefficient of variations (CVs) ≤ 8.5%. Dickkopf-1 (DKK-1), Sclerostin, RANK-L, and OPG were measured by Human Immunoassay ELISA, Biomedica Medizinprodukte GmbH, with intra- and inter-assay CVs, <3%, ≤10%, ≤4%, and ≤5%, respectively. Periostin was measured by Periostin ELISA kit, CLOUD-CLONE CORP., with intra- and inter-assay CVs < 12%. N-terminal-propeptide-procollagen-type-1 (P1NP) was measured by P1NP ELISA kit, Abbexa Ltd., with intra- and inter-assay CVs < 10%. C-terminal-peptide-type-of-collagen (CTx) and bone-specific alkaline phosphatase (BAP) were measured by Human Immunoassay ELISA, Immunodiagnostic Systems Holdings Ltd., with intra- and inter-assay CVs ≤ 10.9%.

### 2.4. Power Analysis

A priori power analysis took place aiming to determine the sample size needed for any statistical relationship to be characterized as significant. For that reason, we used the G*Power 3.0.10 calculator, entering the following mentioned parameters: (a) alpha level of 0.05, (b) statistical power at least of <90%, (c) the effect size for the evaluation of the repeated measurements (i.e., the minimum expected difference) was determined average (f = 0.4) due to the variable duration between them and to reduce the probability differences due to chance. The calculator output indicated that by using 10 patients in total, we could ensure power 85% for all the assessments of repeated measurements analysis performed (within five time points for BMD, BTMs, and inflammatory cytokines). The increase in power alongside with the reduction of the number of subjects needed are because the standard deviation is smaller for changes occurring within the same subjects than between different groups.

## 3. Statistical Analysis

To investigate the effect of HDIST on BMD, BTMs and inflammatory cytokines at different time points (short-term and up to two years), we performed repeated measurements analysis of variance (ANOVA) for normally distributed variables or Friedman ANOVA for those that were not. In addition, multiple pair wise comparison tests, Bonferroni post hoc tests or Wilcoxon matched-pair tests, were applied, respectively, to define between which time points the difference was significant.

Because of the longitudinal design of the study and the inter individual variability of time intervals through which values were assessed, data were expressed as the mean percentage of change from baseline. Relationships between changes in BTMs and other biochemical variables (absolute and percent change from baseline) at different time points were explored using Pearson’s or Spearman’s rank correlation coefficients depending on their distribution (r with 95% confidence interval and *p* values reported). Descriptive statistics for baseline and 24-months characteristics are presented as mean and standard deviation (SD) for each variable. The normal distribution of values for different parameters was verified with the Shapiro–Wilk test which is the appropriate method for small sample sizes. Results for all tests were considered statistically significant at the level of 0.05. All analyses were conducted using SPSS (version 24, IBM Corp, SPSS, Chicago, IL, USA).

## 4. Results

Clinical characteristics and biochemical indices mean values of the 10 enrolled subjects (6 women/4 men) at baseline are illustrated in Table 1. All patients were euthyroid and eugonadal. Mean age was 35.2 ± 8.2 years while all patients were fully ambulatory (EDSS score ≤ 1). Five patients declared themselves social smokers while none declared that they consume alcohol on a regular basis. No significant differences were observed at baseline and in the 24-month follow-up between genders as well as between smokers and non-smokers.

Regarding differences between baseline and 24 months intact parathyroid hormone (iPTH) was the only biochemical marker which showed a statistically significant decrease (mean difference: −8.3 ± 2.11 pg/mL, *p* = 0.021, Table 1). No significant differences were observed for BMD values as shown in Table 2. Similar results regarding BMD values were obtained from repeated measurements analysis. Particularly, a transient non-significant decrease of BMD was observed at all sites 6 months after GC administration, which subsequently appeared to be restored while in the lumbar spine this trend for reduction continued up to 24 months (Figure 2).

Repeated measurements analysis for BTMs and inflammatory cytokines at different time points (short-term and up to two years) revealed no significant differences except from bone formation markers, P1NP (F(4,24) = 4.099, *p* = 0.011) and BAP (F(4,24) = 4.976, *p* = 0.027). P1NP showed an initial non-significant fall on day 6 (mean difference: −0.414 ± 0.128 ng/mL, *p* = 0.181) compared to baseline values, followed by a significant increase on day 90 (mean difference: +0.567 ± 0.13 ng/mL, *p* = 0.048). BAP showed an initial non-significant fall on day 4 (mean difference: −0.864 ± 0.334 μg/L, *p* = 0.416) followed by a significant increase on day 90 (mean difference: +1.838 ± 0.464 μg/L, *p* = 0.05) (Figure 3).

Correlation analysis revealed a positive association of the percentage changes in left femoral neck BMD from baseline to 24 months (−2.73 ± 5.67%) with the percentage changes in Body Mass Index (BMI) (2.73 ± 4.86%), (*r=* +0.661, *p* = 0.038). Percentage changes from baseline to 24 months in left femoral trochanter BMD (1.76 ± 5.41%) were positively correlated with the percentage changes of Periostin (23.93 ± 49.3%), (*r =* +0.77, *p* = 0.009) and negatively with CTx levels (0.89 ± 65.7%), (*r =* −0.782, *p* = 0.008) (Figure 4). The percentage of changes from baseline to 24 months in total left femoral BMD (−2.11 ± 3.03%) were positively correlated with the percentage changes of Periostin levels (*r* = +0.709, *p* = 0.022) (Figure 5). No significant correlations were observed between the percentage changes in lumbar spine BMD (−2.83 ± 6.36%) and the changes of the biochemical indices studied.

Regarding the correlations between biochemical indices, the percentage changes from baseline to 24 months in CTx correlated negatively with the changes in Periostin levels (*r* = −0.806, *p* = 0.005) (Figure 6). Furthermore, the percentage changes of 25OHD levels form baseline to 24 months (11.6 ± 21.19%) correlated negatively with the changes of IL-1β from baseline to 3 (89.9 ± 271.65%) and 6 months (−29.05 ± 33.7%) (*r* = −0.806, *p* = 0.005, and *r* = −0.867, *p* = 0.002, respectively).

## 5. Discussion

The pathophysiology and mechanisms of GC-induced osteoporosis have been studied thoroughly. However, prospective studies evaluating changes of BTMs in relation to BMD under HDIST in MS patients are lacking. In the present study, we have aimed to evaluate such changes focusing on a specific, albeit small group, of relatively homogeneous patient population trying to exclude all possible confounding factors predisposing to osteoporosis.

In our cohort, HDIST did not seem to induce a long-term negative effect on BMD of NDMSP, which is in agreement with previous studies. However, it should be noted that these studies were retrospective, or, despite their prospective design, they had included patients already suffering from MS, not fully ambulatory and of different age groups [29,30,31]. On the other hand, a negative association of BMD with HDIST has been reported in other studies, albeit ambulatory status was one of the major risk factors for reduced BMD [32,33]. In our study, none of the patients was diagnosed with osteoporosis before or during the 24-month follow-up while all of them were fully ambulatory (EDSS ≤ 1), and there were no other osteoporosis risk factors except for smoking, as these were exclusion criteria. The non-significant decrease of BMD was observed at all sites 6 months after GC administration, which subsequently appeared to be restored at all sites except for the lumbar spine, and is possibly explained by the pathophysiology of GC-induced osteoporosis mainly affecting the trabecular bone, which is predominant in the lumbar spine [11,34].

The initial non-significant fall of P1NP, indicating reduced osteoblastic bone formation, is consistent with other studies, as it has been shown that there is direct downregulation of procollagen-α1 (I) gene expression by GCs [29,35,36]. A high bone turnover phase as a “response” to the transient adverse effects of GC administration is documented by the significant increase of bone formation markers, P1NP and BAP, following GC discontinuation on day 90. The significant decrease of iPTH levels from baseline to 24 months is in accordance with the previous finding, indicating a transition to bone formation after a bone resorption phase induced by the inflammatory cytokines involved in MS pathogenesis as well as by the GC effect on bone metabolism [14,15,17,18].

A very interesting finding in our study was the positive correlation between Periostin and femoral BMD. Periostin levels have been associated with increased fracture risk in postmenopausal women “probably suggesting adaptation of the metabolic activity of the periosteum to increased mechanical stress in order to maintain stable bone quality” [37]. Moreover, Periostin levels have been weakly negatively [38] or not at all [39] correlated with femoral BMD of postmenopausal women. On the other hand, changes in Periostin levels—which mainly reflect the metabolic activity of the bone cortex—have shown a positive correlation with changes in femoral BMD of osteoporotic women during teriparatide administration, given the greater ratio of cortical to trabecular bone at that skeletal site [40]. The positive correlation between Periostin and femoral BMD in these patients as well as in our cohort, could perhaps be interpreted through the change in the secretion pattern of PTH—from continuous to intermittent secretion—that has been observed in patients treated with GCs, thus mimicking the teriparatide mechanism of action which is known to stimulate bone formation under intermittent administration [41]; under this aspect, the negative correlation of Periostin with CTx—a bone resorption marker—could be explained. However, it should not be overlooked that our patients were diagnosed with an autoimmune inflammatory disease which, based on observations of Periostin levels in patients diagnosed with diseases of similar pathophysiology, could by itself lead to an increase in Periostin levels [42].

The positive correlation of the changes in left femoral neck BMD with the changes in BMI observed in our cohort is similar to other studies where increased BMI seemed to have a protective effect on bone mass [43,44,45,46]. On the other hand, it has been reported that increased BMI may be specifically protective for certain skeletal sites (i.e, femur), whereas in other sites, it may increase fracture risk (i.e, humerus) [47]. Age, fat mass index (FMI), and lean mass index (LMI) as well as weight status (overweight or obese) are critical parameters mediating the relationship between BMD and BMI. In young individuals, obesity (BMI > 30) may be harmful for bone health, whereas being overweight (BMI > 25 < 29.9) is neutral or protective [48,49] with the lean mass being the most important component for BMD [50]. A bone–adipose axis between adipose tissue with the secretion of adipokines and skeletal tissue with the secretion of osteokines might be a possible mechanism explaining this interplay [51].

Finally, it is worth mentioning the negative correlation between 25OHD and IL-1β, although this correlation was among the percentage of changes of 25OHD levels from the baseline to 24 months and the changes of IL-1β from the baseline to 3 and 6 months. Nevertheless, we should take into consideration that inadequate vitamin D levels have been correlated with higher MS risk and disease activity [52], whereas the role of IL-1β in the pathophysiology of MS is well recognized; in the experimental model of autoimmune encephalomyelitis, autoreactive Th1 and Th17 cells enhance IL-1β production, thus promoting neuroinflammation and CNS damage [53,54]. Moreover, an inverse correlation between vitamin D and inflammatory cytokines has been observed [55] while a down-regulation of the expression and production of cytokines, IL-1β included, has been documented [56].

Our study has strengths and some limitations. We have included a homogenous patient population meeting strict inclusion criteria to eliminate any possible confounding factors. Except from social smoking, which is considered an osteoporosis risk factor, only DMT was not taken under consideration during the analysis. Patient’s young age, disease-free medical history, and long-term follow-up are included in our study’s strengths. Regarding the limitations, the most obvious is that of the small sample size. This study was further limited by the lack of control group to compare changes in BMD and BTMs with age–sex matched controls. 

## 6. Conclusions

Despite the small sample size and the lack of a control group, this is the first prospective study aiming to elucidate the impact of HDIST on BMD and simultaneously on biochemical parameters of bone metabolism in newly diagnosed MS patients. It appears that high dose short term GC administration seems to have no long-term negative effect on BMD in this group of patients. The observed transient increase in bone formation markers —90 days after GC administration—probably indicates a high bone turnover phase as a “response” to the initial adverse effects of GCs on bone metabolism. Additional prospective studies with larger sample size on similarly selected patients should be performed, towards therapeutic decision making and early preventing measures against putative osteoporosis, according to specific personal demographic or disease characteristics of NDMSP.

## Figures and Tables

**Figure 1 biomedicines-11-00603-f001:**
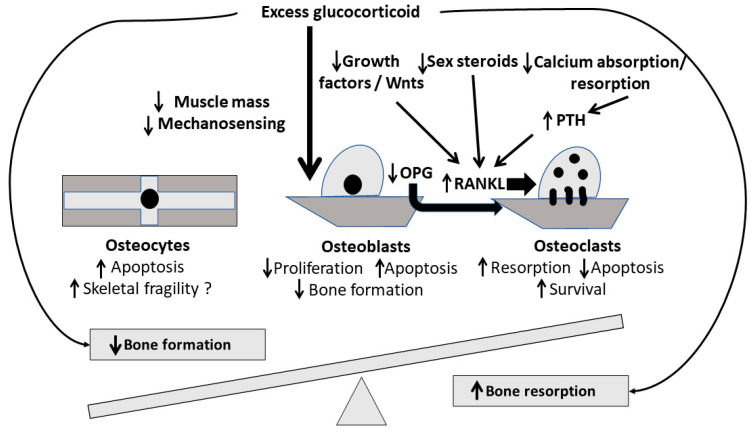
An illustrative overview of GC action on bone remodeling.

**Figure 2 biomedicines-11-00603-f002:**
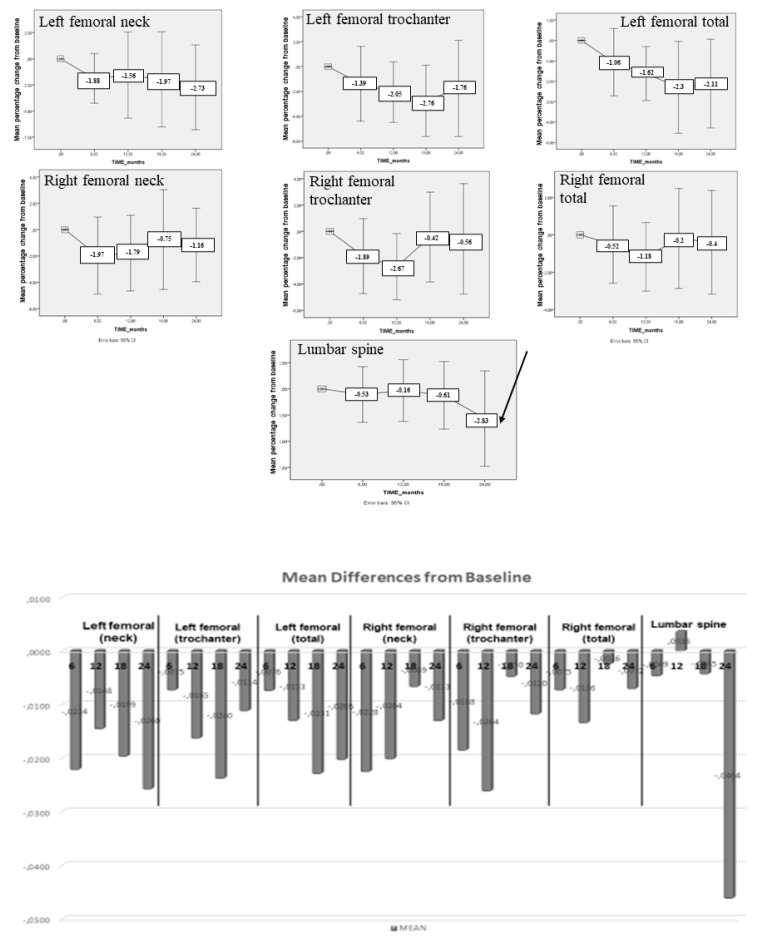
A transient non-significant decrease of BMD was observed at all sites 6 months after GC administration, which subsequently appeared to be restored while in the lumbar spine this trend for reduction continued up to 24 months.

**Figure 3 biomedicines-11-00603-f003:**
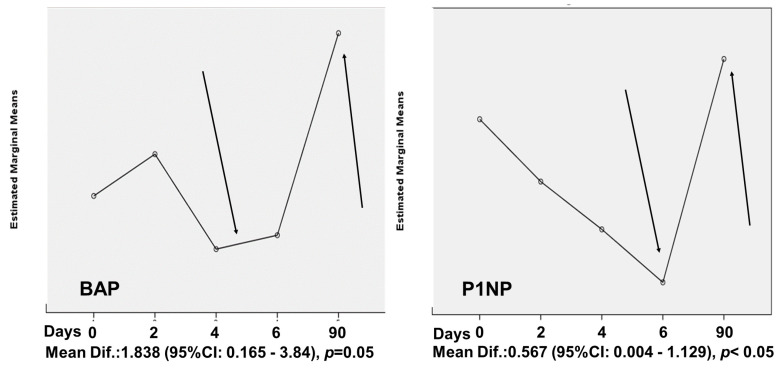
Bone formation markers showed an initial non - significant fall followed by a significant increase on day 90.

**Figure 4 biomedicines-11-00603-f004:**
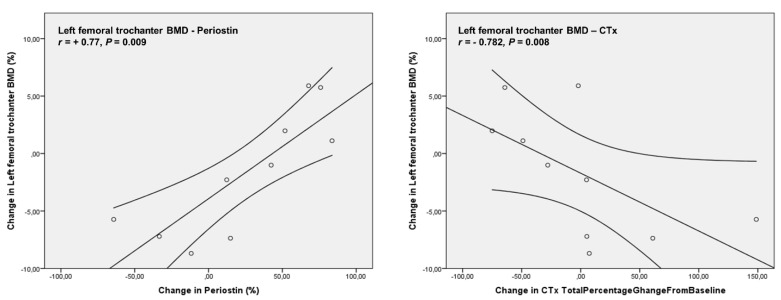
Percentage changes from baseline to 24 months in left femoral trochanter BMD were positively correlated with the percentage changes of Periostin and negatively with CTx levels.

**Figure 5 biomedicines-11-00603-f005:**
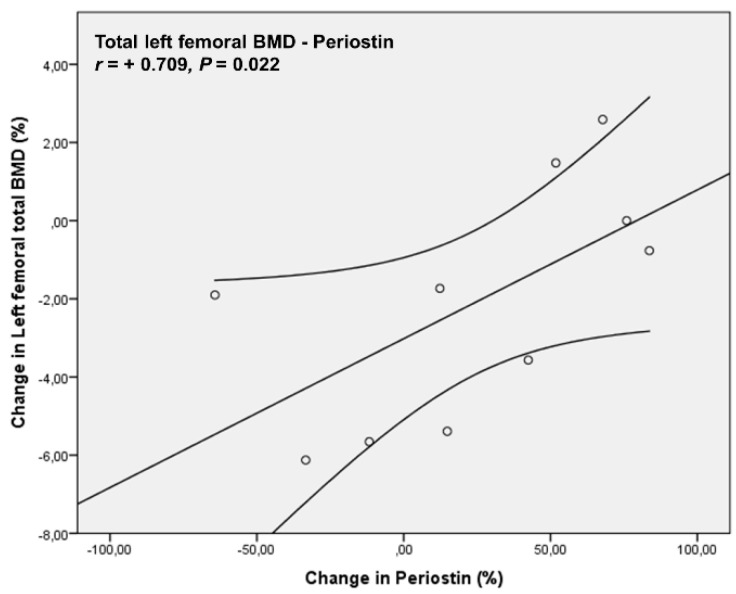
Percentage changes from baseline to 24 months in total left femoral BMD were positively correlated with the percentage changes of Periostin levels.

**Figure 6 biomedicines-11-00603-f006:**
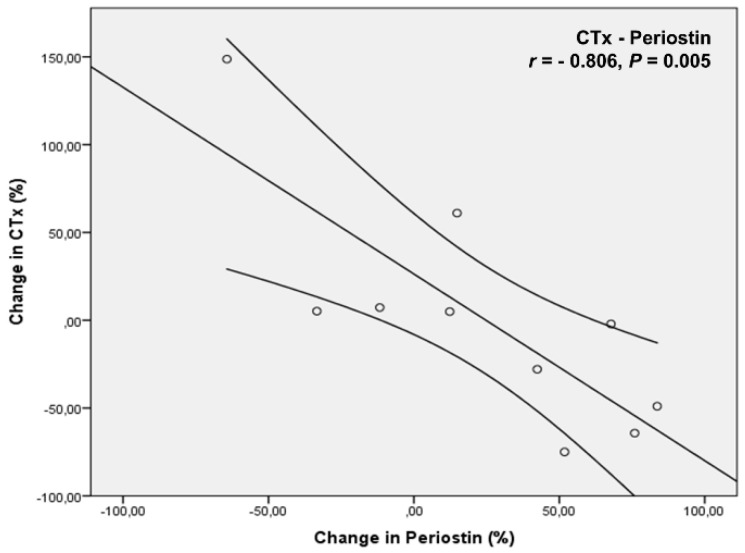
Percentage changes from baseline to 24 months in CTx correlated negatively with the changes in Periostin levels.

**Table 1 biomedicines-11-00603-t001:** Patients’ characteristics at baseline and after 24-months follow-up.

	Baseline Mean ± SD	24-Month f-up Mean ± SD	*p*
Weight (kg)	85.61 ± 13.08	87.86 ± 13.42	0.120
BMI (kg/m^2^)	26.72 ± 2.76	27.43 ± 2.76	0.105
FFM (kg)	56.23 ± 13.27	56.03 ± 12.50	0.770
FAT (kg)	29.37 ± 9.17	31.82 ± 8.96	0.099
TSH (mIU/mL)	1.97 ± 0.49	1.92 ± 0.45	0.837
FT4 (pmol/L)	16.74 ± 2.30	16.94 ± 2.13	0.811
FSH (mIU/mL)	4.35 ± 1.89	7.28 ± 6.66	0.251
LH (mIU/mL)	4.65 ± 1.21	4.70 ± 2.06	0.959
PRL (ng/mL)	8.95 ± 7.16	5.75 ± 2.86	0.72
E2 (pmol/L)	272.45 ± 269.01	157.33 ± 124.17	0.447
Testo (ng/mL)	3.45 ± 4.03	3.82 ± 4.82	0.852
IL1β (pg/mL)	1.50 ± 0.71	1.56 ± 0.69	0.831
IL6 (pg/mL)	2.48 ± 1.05	2.54 ± 1.20	0.900
IL17 (pg/mL)	6.14 ± 2.51	6.57 ± 3.32	0.696
Ca (mg/dL)	9.19 ± 0.31	8.86 ± 0.37	0.060
Alb (g/dL)	4.30± 0.28	4.46 ± 0.19	0.261
*p* (mg/dL)	3.31 ± 0.54	3.49 ± 0.70	0.408
Mg (mg/dL)	2.09 ± 0.31	2.24 ± 0.13	0.223
Cr (mg/dL)	0.86 ± 0.17	0.91 ± 0.20	0.179
25OHD (ng/mL)	30.00 ± 5.60	33.07 ± 6.60	0.137
iPTH (pg/mL)	37.20 ± 15.13	28.90 ± 13.02	**0.021**
CTx (ng/mL)	0.58 ± 0.40	0.42 ± 0.15	0.245
BAP (μg/mL)	7.24 ± 3.62	6.81 ± 2.13	0.718
P1NP (ng/mL)	1.34 ± 0.86	1.31 ± 0.81	0.895
RANKL (pmol/L)	0.23 ± 0.20	0.19 ± 0.16	0.566
OPG (pmol/L)	3.14 ± 1.12	3.32 ± 1.39	0.512
SCL (pmol/L)	37.71 ± 12.53	32.97 ± 9.50	0.438
DKK1 (pmol/L)	19.15 ± 14.24	17.89 ± 11.84	0.756
Periostin (ng/mL)	1695.63 ± 633.03	1906.90 ± 667.80	0.460

**Table 2 biomedicines-11-00603-t002:** BMD values at baseline and after the 24-month follow-up.

BMD	Baseline Mean ± SD (g/cm²)	24-month f-up Mean ± SD (g/cm²)	*p*
Left femoral (neck)	0.95 ± 0.08	0.92 ± 0.08	0.148
Left femoral (trochanter)	0.82 ± 0.10	0.80 ± 0.10	0.327
Left femoral (total)	0.99 ± 0.08	0.97 ± 0.09	0.064
Right femoral (neck)	0.96 ± 0.10	0.95 ± 0.09	0.369
Right femoral (trochanter)	0.82 ± 0.11	0.81 ± 0.12	0.726
Right femoral (total)	0.98 ± 0.10	0.98 ± 0.10	0.738
Lumbar spine	1.24 ± 0.14	1.21 ± 0.16	0.204

## Data Availability

Data is contained within the article.

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
