# Peer review of "High-Dose Intravenous Steroid Treatment Seems to Have No Long-Term Negative Effect on Bone Mineral Density of Young and Newly Diagnosed Multiple Sclerosis Patients: A Pilot Study"

_biomedicines, 2023, doi:10.3390/biomedicines11020603_

Round 1

Reviewer 1 Report

Thank you very much for allowing me to review the article entitled “_High Dose Intravenous Steroid Treatment Seems to Have no Long-Term Negative Effect on Bone Mineral Density of Young and” (biomedicines-2215412).

The aims of this study were, to prospectively investigate the effect of high-dose intravenous steroid treatment on bone metabolism in newly diagnosed multiple sclerosis patients and to evaluate the relationship between bone mineral density and bone metabolism, and inflammatory cytokines.

The introduction is well planned, they must consider that the first time a full word is written it must be indicated including the acronym in parentheses, in the objectives section some acronyms have not been previously explained. Figure 1 is illustrative.

Material and methods should indicate the design used, and if the sample size has been calculated. It is probably a series of cases in which description and analysis are combined, but it is still a series of cases, which is important when considering future studies. Please clarify this situation as it is of great importance to assess the scientific evidence of the results.

please check the sample size calculation or explain it further.

How have you assessed the normality of the distribution?

In the discussion the authors say that they have controlled for potential confounders but have not really used any multivariate model that can adjust for these factors.

Reviewer 2 Report

We read the article titled High Dose Intravenous Steroid Treatment Seems to Have no Long-Term Negative Effect on Bone Mineral Density of Young and Newly Diagnosed Multiple Sclerosis Patients by  Simeakis et where the authors evaluate the relationship between Chronic oral glucocorticoid (GC) administration correlates with bone loss in multiple sclerosis patients. There are a number of comments; the study lacks any power analysis that substantiates the results:

starting with 25 patients and ending in 10 patients, would render this work "possibly nonsignificant" and the results may be occurring by chance alone.

I would suggest that the title be changed to a Pilot study,

second, statistical analysis and power analysis should be performed to see if there is any validity in the results

the work did not include any controls, maybe patients of similar age and sex that are not being administered GC.

A copy of the consent form should be included in the supplementary data.,

Round 2

Reviewer 1 Report

In connection with the submitted article “High Dose Intravenous Steroid Treatment Seems to Have no Long-Term Negative Effect on Bone Mineral Density of Young and” (biomedicines-2215412), I have carefully reviewed the new version of the manuscript as well as the authors’ answer. I consider your response in relation modifications made are appropriate to the suggestions made.

Reviewer 2 Report

accept